# Length Effect on the Stress Detection of Prestressed Steel Strands Based on Electromagnetic Oscillation Method

**DOI:** 10.3390/s19122782

**Published:** 2019-06-20

**Authors:** Benniu Zhang, Chong Tu, Xingxing Li, Hongmei Cui, Gang Zheng

**Affiliations:** 1School of Civil Engineering, Chongqing Jiaotong University, Chongqing 400074, China; tu_chong@foxmail.com (C.T.); zhenggang@cmhk.com (G.Z.); 2School of Information Science and Engineering, Chongqing Jiaotong University, Chongqing 400074, China; xingxingli331@cqjtu.edu.cn; 3Chongqing Telecommunication Polytechnic College, Chongqing 402247, China; cui_hongmei@foxmail.com

**Keywords:** length effect, stress detection, electromagnetic oscillation, steel strand, concrete structures

## Abstract

Prestress detection of structures has been puzzling structural engineers for a long time. The inductance–capacitance (LC) electromagnetic oscillation method has shown a potential solution to this problem. It connects the two ends of a steel strand, which is simulated as an inductor, to the oscillation circuit, and the stress of the steel strand can be calculated by measuring the oscillation frequency of the circuit through a frequency meter. In the previous studies, the authors found that stress-frequency relation of 1.2 m steel strand was negatively correlated, while the stress-frequency of 10 m steel strand was positively correlated. To verify this conflict, two kinds of electrical inductance models of steel strands were established to fit the lengths. With the models, the stress-frequency relations of steel strands with different lengths were analyzed. After that, two kinds of experimental platforms were set up, and a series of stress-frequency relationship tests were carried out with 1.2 m, 5 m, 10 m and 15 m steel strands. Theoretical analysis and experimental results show that when the length is less than 2.013 m, the stress and oscillation frequencies are negatively correlated; when length is more than 2.199 m, the stress and oscillation frequencies are positively correlated; while when length is between 2.013 m and 2.199 m, the stress-frequency relationship is in transit from negative correlation to positive correlation.

## 1. Introduction

Compared with ordinary reinforced concrete structures, prestressed concrete structures have a lighter weight, smaller cross-section size, and better economy. The determination of the prestress loss is an important criterion for judging the bearing capacity of the prestressed structure. However, the prestress loss caused by the shrinkage and creep of the concrete and the relaxation of the prestressed steel strand directly threatens the safety of the structure. Therefore, it is particularly important to detect the effective prestressing force in the existing structure. The current prestress detection research includes the following.

### 1.1. Ultrasound Guided Wave Method

The ultrasonic guided wave method uses the penetration and propagation of ultrasonic waves to achieve stress detection. In recent years, ultrasonic guided waves have made some progress in the field of concrete structural stress nondestructive detection. For example, Qian et al. [1] verified the effect of ultrasonic guided wave energy entropy spectroscopy on the stress of 7-wire steel strands by numerical simulation and a series of tests. Niederleithinger et al. [2] used a network of 20 ultrasonic transducers to qualitatively detect the stress state of concrete beams. Feng et al. [3] designed an instrument with an excitation and receiving frequency greater than 100 kHz for steel strand damage detection based on the principle of the ultrasonic guided wave. The device can identify the frequency change caused by the groove with a cross-section loss of 1.13%. Xu et al. [4] used the ultrasonic guided wave below 400 kHz to realize the detection of steel strand breakage defects. Farhidzadeh et al. [5,6,7] achieved some success.

### 1.2. Embedded Fiber Sensors or Magnetoelastic Devices

Embedded fiber optic sensors use the change of optical signal to monitor the stress of structures. This type of technology has achieved some significant results in the field of civil engineering [8,9,10]. Typically, Huynh et al. [11] embedded optical fiber sensors in unbonded prestressed concrete beams to monitor the stress loss of prestressing tendons under temperature. Kim et al. [12] used fiber optic sensors to monitor the stress variation of the 60 m span beam during the whole construction process. Compared with the traditional Bragg grating, the advantage is that these sensors can still effectively monitor the strain after the concrete has begun cracking. In addition, Lan et al. [13] made some great progress in the research of fiber optic sensors to monitor the stress of steel strand.

The magneto-elastic devices, which are based on the magneto-elasticity method, utilize the characteristics that the ferromagnetic material changes magnetically under mechanical stress to perform stress monitoring. As a typical ferromagnetic material [14], steel strands have a corresponding relationship between stress and magnetic properties. The magneto-elastic effect is mainly used in cables, stress detection of unbonded steel strands [15,16], and prestress monitoring of new structures with a bonded steel strand [17,18,19]. Usually, the excitation coil emits the excitation magnetic field, and the receiving coil receives the magnetic field passing through the material. According to the relationship between the output voltage and the permeability of the material, the corresponding stress of the cable under a certain voltage is obtained. Yim at al. [20] developed a set of stress devices by using the magnetostrictive inverse effect of ferromagnetic materials, which was applied to stress monitoring of bridge cables. By comparing the numerical analysis and monitoring results, the effectiveness of the electromagnetic device in cable force monitoring was verified.

### 1.3. Electromagnetic Oscillation Method

Inspired by the magneto-elastic effect, the author proposed a prestressed steel strand stress detection method based on LC electromagnetic oscillation [21,22], indirectly measuring the strand stress through the electromagnetic oscillation frequency. According to the electrical theory, prestressed steel strands exhibit single conductance properties in low frequency oscillating circuits and can be divided into inductors, capacitors and resistors. When the steel strand is tensioned axially, the rate of inductance change is much larger than that of capacitance and resistance. Therefore, based on the LC electromagnetic oscillation to measure the stress of the prestressing strand, the capacitance and resistance can be neglected, and the steel strand can be modeled as an inductance. In the stress detection process, only the two ends of the strand were connected to the oscillating circuit. The axial tensile force was applied to the steel strand, and the stress signal of the prestressed steel strand was converted into a circuit oscillation frequency signal, and the stress value of the steel strand can be calculated by collecting the oscillation frequency by the frequency meter.

In the previous study [21], the theoretical model of the stress and oscillation frequencies of a steel strand was established, and the axial tensile test was carried out with six sets of 7-wire prestressed steel strands with a length of 1.2 m. This study was the first time that the LC electromagnetic oscillation method had been applied to the stress measurement of a steel strand by the authors, and the feasibility of this method was proved. The theoretical analysis and experimental results of the study show that the circuit oscillation frequency decreases with the increase of steel strand stress. In another study [22], Taylor’s expansion equation was used to establish the relationship between steel strand stress and oscillation frequency, and a 7-wire prestressed steel strand with a length of 10 m was used for cyclic loading experiments. The research results show that the oscillation frequency of the circuit increases with the increase of stress.

The above studies find that the stress and oscillation frequency of the two different lengths of steel strands show opposite trends. In the case of using the same oscillating circuit and measurement method, the relationship between the stress of the 1.2 m and 10 m steel strands and the oscillation frequency is completely opposite. The results of previous studies [21,22] cannot explain the reasons for this phenomenon. Only the force–frequency variation trend of 1.2 m and 10 m steel strands were studied, and the feasibility of the inductance–capacitance (LC) electromagnetic oscillation method was verified, while the reasons for the positive (negative) correlation of force–frequency variation were not explained. According to the author’s preliminary analysis, this phenomenon is directly related to the length of steel strand. Therefore, this paper has carried out related theoretical and experimental research on the length effect of the LC electromagnetic oscillation method for the stress detection of steel strands.

## 2. Theories and Models

According to the electrical theory, as part of the LC low-frequency electromagnetic oscillating circuit, the axial stretching of the strand is mainly represented by the change of inductance. The possible reasons for the opposite stress-frequency relationship between different lengths of steel strands can be described by the following two aspects: (a) when the length is short, the spiral characteristics of the seven steel wires are not obvious, but are similar to the segment wire, so it mainly shows the inductance characteristics of the segment wire; (b) when a certain length is reached, the spiral characteristics of seven steel wires have been formed. The steel strand approximates the spiral coil structure, and mainly exhibits the spiral coil inductance characteristics.

Different length steel strands show different inductance characteristics. Therefore, in order to further study the influence of the relationship between stress and oscillation frequency, the author establishes the mathematical models of stress-frequency under two inductance models based on the model of conductor inductance and the model of spiral coil inductance and the principle of LC electromagnetic oscillation.

### 2.1. Principle of LC Electromagnetic Oscillation

If a steel strand under axial tension is regarded as having the equivalent inductance of an LC oscillating circuit, its magnetic and inductance parameters will change when stretched, and then the internal stress of the steel strand can be expressed by frequency through the oscillating relationship between inductance and capacitance in the electromagnetic oscillating circuit.

As shown in Figure 1, the LC oscillating circuit, also be called the LC resonant circuit, which consists of an inductor and a capacitor connected together. A steel strand is used instead of the inductor to connect to the oscillating circuit, and a fixed capacitor component is externally connected to form an LC oscillating circuit. The basic working principle can be described as follows: (a) when the capacitor starts to discharge, the inductor is charged. When the voltage of the inductor reaches the maximum, the capacitor is discharged and the inductor is charged; (b) the inductor begins to discharge, the capacitor is charged, and when the capacitor is charged, the inductor is discharged and the capacitor is charged. In this reciprocating operation, resonance occurs.

It is assumed that the circuit has no energy loss and is not affected by other external factors. When the circuit resonates, the inductive reactance $X_{L}$ of the inductor *L* in the oscillating circuit is equal to the capacitive reactance $X_{C}$ of the capacitor *C*.

(1)$$X_{L}=X_{C}$$

The inductance $X_{L}$ of the inductor can be expressed as

(2)$$X_{L}=2\pi fL$$

The reactance $X_{C}$ of the capacitor can be expressed as

(3)$$X_{C}=\frac{1}{2\pi fC}$$

Therefore, the relationship between the oscillation frequency of the circuit and the inductance of steel strand can be obtained by combining Equations (1)–(3).

(4)$$2\pi fL=\frac{1}{2\pi fC}$$

The oscillation frequency of the circuit can be expressed as

(5)$$f=\frac{1}{2\pi \sqrt{LC}}$$

In the equation, f is the oscillation frequency of the circuit, *L* is the inductance of the steel strand, and *C* is the capacitance of the circuit.

*C* is a constant, and the circuit oscillation frequency f can be directly measured by the frequency. Therefore, it is only necessary to establish the relationship between the inductance *L* and the stress σ, and the relationship model between the stress of the steel strand and the oscillation frequency of the circuit can be obtained.

### 2.2. Inductance Model of Segment Wire

When the steel strand is short, the spiral structure has not yet been formed, and its structural characteristics are similar to those of the line segment, so that the inductance characteristics of the segment wire are mainly exhibited. In this section, the model of stress-frequency of the steel strand is established based on the model of inductance of the segment wire.

#### 2.2.1. Modeling

When a conductor passes a current, a magnetic field is generated inside and around the conductor. The magnetic circuit is a concentric closed magnetic field ring, and the direction is determined by the right hand rule. When the current changes, the magnetic linkage changes accordingly. For non-magnetic materials, the inductance of the conductor is the ratio of its total flux chain to the current, also known as static inductance.

(6)$$L=\frac{\psi }{I}$$

In the equation, ψ is the self-inductive magnetic linkage of the conductor, *I* is the circuit current, and *L* is the inductance of the conductor.

In the previous study [21], the authors discussed the internal inductance of the wire, and the internal inductance $L_{in}$ can be expressed as

(7)$$L_{in}=\frac{\psi_{in}}{I}=\frac{\mu l}{8\pi }$$

In the equation, $L_{in}$ is the internal inductance of the wire, μ is the permeability of the wire, and the *l* is the length of the wire.

As shown in Figure 2, for the external inductance produced by the wire segment, the external magnetic circuit contains the current of the whole wire. According to the Biot-Savart law, the magnetic induction intensity at any point outside the strand can be obtained.

The Biot-Savart law can be defined as

(8)$$d\vec{B}=\frac{\mu_{0}}{4\pi }\cdot \frac{Id\vec{l}\times \vec{r}}{r^{3}}$$

The magnetic induction intensity dB produced by current element Idy at point P can be expressed as

(9)dB=μ0I4π·dyc2sinθ=μ0I4π·Ixdyx2+y−b232

Hence the magnitude of the magnetic induction intensity scalar produced by the segment wire at point P is as follows:(10)B=μ04π∫0lIxx2+y−b232dy=μ0I4πl−bxx2+l−b2+bxx2+b2

If the wire radius is *r*, the total flux of the wire segment *l* is

(11)$$\Phi_{e}=\int \vec{B}d\vec{S}=\overset{\infty }{\underset{r}{\int }}\overset{l}{\underset{0}{\int }}\frac{\mu_{0}I}{4\pi }\left[\frac{l-b}{x\sqrt{x^{2}+{\left(l-b\right)}^{2}}}+\frac{b}{x\sqrt{x^{2}+b^{2}}}\right]dxdy=\frac{\mu_{0}I}{2\pi }\left(l\cdot ln\frac{l+\sqrt{l^{2}+r^{2}}}{r}-\sqrt{l^{2}+r^{2}}+r\right)$$

All fluxes are only interlinked with a single wire, so the external magnetic linkage is equal to the flux. Which is

(12)$$\Phi_{e}=\psi_{out}$$

Therefore, the external inductance of the wire can be expressed as

(13)$$L_{out}=\frac{\psi_{out}}{I}=\frac{\mu_{0}}{2\pi }\left(l\cdot ln\frac{l+\sqrt{l^{2}+r^{2}}}{r}-\sqrt{l^{2}+r^{2}}+r\right)$$

The inductance of the wire is the sum of the internal inductance and the external inductance. Namely,

(14)$$L=L_{in}+L_{out}=\frac{\mu l}{8\pi }+\frac{\mu_{0}}{2\pi }\left(l\cdot ln\frac{l+\sqrt{l^{2}+r^{2}}}{r}-\sqrt{l^{2}+r^{2}}+r\right)$$

The length of the wire is much larger than the radius, that is, l≫r, the above formula can be simplified to

(15)$$L=\frac{\mu l}{8\pi }+\frac{\mu_{0}l}{2\pi }\left(ln\frac{2l}{r}-1\right)$$

The steel strand can be equivalent to a segment wire. Since the steel wire magnetic permeability μ is much larger than the vacuum magnetic permeability $\mu_{0}$, the upper equation can be simplified to

(16)$$L=\frac{\mu l}{8\pi }$$

The length of the steel strand after stretching can be expressed as

(17)$$l=l_{0}+\Delta l$$

In the equation, $l_{0}$ is the original length of the steel strand, and Δl is the elongation of the steel strand.

From the constitutive relation of materials,

(18)$$\sigma =E\varepsilon =E\frac{\Delta l}{l}$$

The elongation Δl can be expressed as

(19)$$\Delta l=\frac{\sigma l_{0}}{E}$$

In the equation, σ is the stress of the steel strand and E is the elastic modulus of the steel strand.

And the permeability of steel strand can be expressed as

(20)$$\mu =\mu_{0}\mu_{r}$$

Combining (5), (17), (19) and (20), it can be obtained that

(21)f=12πμ0μrl0+σl0E8π·C

#### 2.2.2. Simulation Result

Substitute $\mu_{0}=4\pi \times 10^{-7}\;H\cdot m^{-1}$, $\mu_{r}=1500$, $l_{0}=1.2\;m$, $E=1.95\times 10^{5}\;\mathrm{MPa}$, C=0.0059 uF,  into Equation (21) and it can be obtained that

(22)$$f=\frac{1000}{2\pi \sqrt{5.311+3.63\times 10^{-6}\sigma }}$$

The simulation of the stress-frequency of the 1.2 m steel strand is shown in Figure 3.

According to the stress-frequency relationship of the strand and Figure 3, it can be obtained that the oscillation frequency of the circuit decreases with the increase of stress. This phenomenon coincides with the author’s study on the stress and frequency of a 1.2 m steel strand [21]. Therefore, it can be inferred that the 1.2 m steel strand mainly shows the inductance characteristics of the segment wire.

### 2.3. Inductance Model of Spiral Coil

When the steel strand reaches a certain length, the steel wire has formed a spiral structure, which is similar to the spiral coil and mainly shows the inductive characteristics of the spiral coil. It can be deduced from electrical theory that the inductance of a straight wire with a certain length is much smaller than the inductance of a spiral coil wound by a straight wire with the same length. Therefore, the inductance effect of the straight wire of the long steel strand can be neglected compared to the inductance effect of the spiral coil. Based on the spiral coil inductance model, this section establishes a relationship model between the stress and oscillation frequency of the long steel strand.

#### 2.3.1. Modeling

As shown in Figure 4, when the steel strand is characterized by the spiral coil inductance, the peripheral steel wire can be regarded as a spiral coil, and the steel wire at the center of the cross section of the strand can be regarded as the core of the spiral coil.

The total magnetic linkage of the spiral coil can be expressed as

(23)ψ=μHNS

In the equation, ψ is the total magnetic linkage of the spiral coil, μ is the magnetic permeability of the magnetic core in the spiral coil, H is the magnetic field intensity, N is the number of turns of the spiral coil, and S is the cross-sectional area of the spiral coil.

For a spiral coil, the magnetic field intensity can be defined as

(24)$$H=\frac{NI}{l}$$

In the equation, I is the current of the spiral coil, and l is the length of the spiral coil.

Combining Equations (6), (23) and (24), it can be obtained that

(25)$$L=\frac{\mu N^{2}S}{l}$$

The cross-sectional area and length of the spiral coil will change as the steel strand is stretched. Therefore, it is necessary to calculate the amount of change of *S* and *l* after steel strand is elongated.

For the stretched steel strand, the area of coil cross-sectional can be approximately expressed as

(26)$$S=\pi R^{2}=\pi \left(R_{0}-\Delta R\right)$$

In the equation, $R_{0}$ is the initial section radius of the steel strand, and ΔR is the amount of change of radius.

By combining Equations (17), (19), (25) and (26), and ignoring high order item $\Delta R^{2}$, it can be obtained that

(27)L=μπN2(R02+R0·R)l0+σl0E

Since $R_{0}\gg \Delta R$, Equation (27) can be simplified to

(28)L=μπN2R02l0+σl0E

By combining Equations (5), (20) and (28), it can be obtained that

(29)f=12πμ0μr·πN2R02l0+σl0E·C

#### 2.3.2. Simulation Result

Substitute $\mu_{0}=4\pi \times 10^{-7}\;H\cdot m^{-1}$, $\mu_{r}=1500$, N=47, $R_{0}=0.01524\;m$, $l_{0}=10\;m$, $E=1.95\times 10^{5}\;\mathrm{MPa}$, C=0.0059 uF into Equation (29) and it can be contained that

(30)$$f=\frac{\sqrt{1.95\times 10^{5}+\sigma }}{0.439\pi^{2}}$$

The stress-frequency simulation result of the 10 m steel strand is shown in Figure 5.

It can be seen from the relationship between the stress and frequency of the strand and Figure 5 that the oscillation frequency of the circuit increases with the increase of the stress. This phenomenon is consistent with the author’s research on the stress and frequency variation of the 10 m steel strand [22]. Therefore, it can be inferred that the 10 m steel strand mainly exhibits the spiral coil inductance characteristics.

In summary, the steel strand exhibits different inductance characteristics in the LC electromagnetic oscillation circuit, and the length of steel strand is the main factor. Through the author’s previous research [21,22] and the two-inductance model based on the steel strand stress-frequency model, it can be seen that different lengths of steel strands exhibit different inductance characteristics, showing different force-frequency relationships under LC electromagnetic oscillation. Namely, it can be seen that, (a) when the strand is short, it can be modeled as a inductance model of segment wire. After the axial tensile stress is applied to the steel strand, the inductance becomes large, and the relationship between the stress and the oscillation frequency is such that the stress increases and the oscillation frequency decreases; and (b) when the strand reaches a certain length, it can be modeled as a inductance model of spiral coil. After the axial tensile stress is applied to the steel strand, the inductance is reduced, and the relationship between stress and oscillation frequency is such that the stress increases and the oscillation frequency also increases.

## 3. Experimental Studies

### 3.1. Parameters of Steel Strand

In order to analyze the inductance effect model of steel strands in a practical application, steel strands with lengths of 1.2 m, 5 m, 10 m and 15 m were selected as experimental objects in combination with the test conditions, and the parameters are shown in Table 1.

### 3.2. Experimental Systems

#### 3.2.1. Experimental Devices and Procedure of Short Steel Strand (1.2 m)

The experimental system is presented in Figure 6. The steel strand, the frequency meter, and the LC oscillating circuit were connected in series to form a closed circuit, and the oscillating circuit and the frequency meter were supplied by the DC power. The hydraulic universal testing machine applied tension to the steel strand, and the stress and strain data was collected by the computer during the loading process. The accuracy of the frequency meter in the circuit is 0.0001 kHz.

The experimental process is as follows: firstly, open the material software of hydraulic universal testing machine, set parameters such as cross-sectional area of steel strand and tension rate applied in tension experiment. Then, turn on the power supply, control the universal testing machine to preload steel strand to 2 kN, and check whether the frequency meter and LC oscillation circuit are working properly. After confirming that the applied load and the strand deformation tend to be stable, uniformly apply the load to 8 kN at a loading rate of 10 mm/min.

During the loading process, it is necessary to observe the deformation of the steel strand and the accuracy of the frequency data collected by the frequency meter. The frequency meter collects frequency data once per second, and the collected data is transmitted to the mobile device via Bluetooth. Stop the loading when the load reaches 8 kN, and a set of stress and oscillation frequency test data will have been collected in one test. After the data acquisition is completed, the universal testing machine is controlled to unload the tensile force applied to the steel strand to zero while observing the trend of the frequency. A total of four sets of data were collected in the experiments.

#### 3.2.2. Experimental Devices and Procedure of Long Steel Strand (5 m, 10 m and 15 m)

In the construction of the test system, the steel strand and hydraulic jack should be fixed on the steel reaction frame to ensure the normal follow-up work. Then, choose two areas near the middle span of the strand, grind the surface protective layer and rust with sandpaper, and paste the strain gauge. Finally, weld the wire of the circuit at the two free ends of the strand, and ensure the lead-out wire is connected to the LC oscillation circuit. Finally, connect the strain acquisition instrument, frequency acquisition instrument and oscillation circuit to the experimental system. The experimental framework is shown in Figure 7.

Before starting the experiment, it is necessary to check the wire connection at both ends of the steel strand, and confirm that the two welded ends are wrapped with insulating tape after the welding is firm, so as to avoid separation between the two during the experiment. In addition, the data line connecting the strain gauges needs to be fixed on the steel strands, because the steel strands will rotate and drive the data lines, which will affect the connection of the data lines and the strain gauges.

After the above steps are completed, turn on the circuit to check the connection of each part and whether the circuit was oscillating. The stress and frequency test acquisition is carried out by controlling the loading and unloading of the piercing hydraulic jack to ensure the normal operation of the strain acquisition instrument and frequency acquisition device. Frequency meter and static strain acquisition instrument are set up once a second.

The stress and frequency data were collected when the stress of the steel strand reached 200 MPa, and the maximum stress applied was 900 MPa. The hydraulic jack is preloaded to 200 MPa, and the initial frequency data is collected and recorded. The stress was unloaded when loaded to 900 MPa, and the corresponding frequency data was recorded up to 200 MPa per 100 MPa of unloading. A total of four sets of data were collected in the cyclic loading experiments.

### 3.3. Experimental Data

In order to study the relationship between the stress and oscillation frequency of different lengths of steel strands, the median filter was used to process the frequency and stress data which were obtained under the same conditions, and the standard deviation and repeatability error of the oscillation frequency of steel strands with different lengths was calculated. Table 2, Table 3, Table 4 and Table 5 show the test data of the 1.2 m, 5 m, 10 m and 15 m steel strands. Figure 8, Figure 9, Figure 10 and Figure 11 show the fitting curves obtained from the test data of steel strands.

The following conclusions can be drawn from the analysis:(1)The repeatability error of the 1.2 m steel strand test data does not exceed 0.023%, and that of the 5 m, 10 m and 15 m steel strands test data are less than 0.025%, 0.05% and 0.05%.(2)Different lengths of steel strands have different stress-frequency trends. The data analysis of the tensile test of the 1.2 m steel strand shows that the frequency decreases with the increase of stress; while the test analysis of 5 m, 10 m and 15 m steel strands shows that the frequency increases with the increase of stress. It can be seen that the 1.2 m steel strands mainly exhibit the inductance characteristics of the segment wire, while the 5 m, 10 m and 15 m steel strands exhibit the inductance characteristics of the spiral coil.(3)The linearity of stress and frequency fitting curves of different lengths of strands is diversity. The correlation of the stress-frequency fitting curves of the 5 m, 10 m and 15 m strands are 0.8569, 0.9221 and 0.9801. With the increase in the length of the steel strand, the more concentrated the experimental data of stress-frequency is, and the better linear correlation of the curves.(4)In summary, the steel strands exhibit different inductance characteristics in the LC electromagnetic oscillation circuit, and the length of steel strand is the main factor.

## 4. Results and Discussion

The theoretical derivation and experimental analysis results show that length is the critical factor affecting the stress detection of steel strand in electromagnetic resonance method. Within different length intervals, steel strands show different inductance characteristics, which make the force–frequency curve show positive and negative correlation. In order to confirm the length interval of wire inductance or spiral coil inductance of steel strand, this section is based on experimental data and the stress-frequency mathematical model to analyze the length effect of experiment and simulation.

### 4.1. Analysis of Length Effect

In order to obtain the critical length of the long and short steel strands determined by the experiment, four sets of experimental data of different lengths were fitted to get the force-frequency relationship curves. The experimental data were normalized and analyzed to compare the stress-frequency variation of steel strands with different lengths. The normalized stress-frequency curves are shown in Figure 12.

It can be concluded from the figure that:(1)The variation trend of stress-frequency curve of the four steel strands are different. The stress-frequency of the 1.2 m steel strand is negatively correlated, while the other three length are positively correlated.(2)The stress-frequency variation trends of 1.2 m and 10 m steel strands coincided with the results of the author’s previous research. What is more, the force–frequency curve of the 1.2 m steel strand shows large dispersion, and the general variation trend is that the oscillation frequency decreases with the increase of the stress. Compared with the 1.2 m steel strand, the stress-frequency curves of 5 m, 10 m and 15 m steel strands have smaller dispersion and better repeatability. The general variation trend is that the resonant frequency increases with the increase of stress.(3)The stress-frequency variation rule of steel strands transitions from a negative correlation of 1.2 m to a positive correlation of 5 m, indicating that the critical length of the long and short steel strands is within the range of (1.2 m, 5 m). Therefore, the critical length can be obtained by fitting the relationship between f and σ in the stress-frequency fitting curve obtained from the experimental data of each length.

The critical length is related to the rate of change of the force frequency. Therefore, the length l and frequency f of the four kinds of steel strands and the slope $k_{\left(f/\sigma \right)}$ are taken as the data points of the length effect analysis. The fitting curve obtained by fitting the data points can determine the critical length. The length l of the strand has a corresponding relationship with the oscillation frequency f and the stress σ in the fitting curve equation, and the relationship can be described as a two-dimensional length effect point. Therefore, four different length steel strands correspond to four two-dimensional length effect points, and the relationship between l and $k_{\left(f/\sigma \right)}$ can be obtained by fitting. The two-dimensional length effect points obtained from the experimental data are (1.2, -0.000143), (5, 0.000174), (10, 0.000262) and (15, 0.000359). Thus, the experimental length effect curve equation can be expressed as

(31)$$k_{\left(f/\sigma \right)}=1.52\times 10^{-4}\ln\left(l-0.546\right)-5.828\times 10^{-5}$$

Similar to the experimental length effect, the simulation critical length is obtained by the relationship between f and σ in the stress-frequency fitting curve of each length. From the stress-frequency mathematical models established in the Section 2, four sets of simulated two-dimensional length effect points are (1.2, –0.000120), (5, 0.000186), (10, 0.000261) and (15, 0.000345). The equation for the simulation length effect curve can be expressed as

(32)$$k_{\left(f/\sigma \right)}=1.27\times 10^{-4}\ln\left(l-0.899\right)-3.335\times 10^{-5}$$

The experimental length effect curve and the simulation length effect curve are shown in Figure 13. In Equations (31) and (32), when $k_{\left(f/\sigma \right)}$ is zero, the critical length of the stress-frequency relationship of steel strand under the electromagnetic oscillation method can be obtained from the negative correlation to the positive correlation.

### 4.2. Analysis and Discussion

Figure 13 shows the relationship between the experimental length effect curve and the simulation length effect curve. According to length effect analysis and Figure 13, the following conclusions can be drawn from the analysis.

(1)The experimental length effect curves and Equation (31) show that the stress-frequency variation trend of steel strand is related to the length. The critical length for distinguishing long or short steel strands is l=2.013 m. When 0<l<2.013 m, the oscillation frequency decreases with the increase of stress; when 2.013 m<l≤15 m, the oscillation frequency increases with the increase of stress.(2)Similar to the analysis results of experimental length effect, the stress-frequency variation trend of simulation length effect is also related to the length of steel strand. The critical length for distinguishing long or short steel strands is l=2.199 m. When 0<l<2.199 m, the oscillation frequency decreases with the increase of stress; when 2.199 m<l≤15 m, the oscillation frequency increases with the increase of stress.(3)The error of critical length between experimental length effect and simulation length effect is 8.46%, which meets the requirement of application.(4)It can be inferred from the trend of the length effect curve that when the length of steel strand is longer than 15 m, the oscillation frequency increases with the increase of stress.

It can be concluded that different length steel strands exhibit different inductance characteristics in stress detection based on LC oscillation. Both theoretical derivation and experimental analysis show that when the length of steel strand is short, the inductance characteristic of a segment wire is mainly exhibited, and the oscillation frequency decreases with the increase of stress. And when a certain length is reached, the steel strand mainly exhibits the inductance characteristics of spiral coil, and the oscillation frequency increases with the increase of stress.

To summarize, when 0<l<2.013 m, the stress is negatively correlated with the oscillation frequency, that is, the oscillation frequency decreases with the increase of stress, and the main performance of steel strand is the inductance of a segment wire. When l>2.199 m, the stress is positively correlated with the oscillation frequency, that is, the oscillation frequency increases with the increase of stress. At this time, the main performance of steel strand is the inductance characteristic of a spiral coil. The transition interval of length effect is 2.013 m≤l≤15 m, that is, the relationship between stress and oscillation frequency changes from a negative correlation to a positive correlation. The effects of length and external magnetic field are considered in this study, but if there are new influencing factors, the accuracy and consistency of the theoretical and experimental results cannot be guaranteed.

The prestress of the concrete bridge is mainly provided by the steel strand. Compared with other types of prestressed concrete bridges, the span of the simply supported girder bridges is the smallest and generally larger than 15 m. When the stress is measured by the LC electromagnetic oscillation method, the steel strand mainly shows the inductive characteristics of spiral coils. Consequently, it is appropriate to choose the stress-frequency mathematical model of steel strands based on the inductance characteristics of spiral coils in engineering applications.

## 5. Conclusions

This paper established the stress-frequency models of long and short steel strands. The theoretical models were validated by stress-frequency experiments of four lengths of 1.2 m, 5 m, 10 m and 15 m, and the inductance characteristics of each length of steel strand were analyzed. Both theoretical analysis and experimental research show that: (a) short steel strands mainly represent the inductance characteristics of the segment wires, the oscillation frequency decreases with the increase of stress; (b) while the long steel strands mainly represent the inductance characteristics of spiral coils, the oscillation frequency increases with stress increases.

The error of critical length obtained from the analysis of experimental length effect and simulation length effect is 8.46%, and the critical length interval for distinguishing long or short steel strands is (2.013 m, 2.199 m). When 0<l<2.013 m, the oscillation frequency decreases with the increase of stress. When l>2.199 m, the oscillation frequency increases with the increase of stress. When 2.013 m≤l≤2.199 m, the transition interval of the length effect of the stress measurement is based on the electromagnetic oscillation method, that is, the stress and the oscillation frequency are transitioned from a negative correlation to a positive correlation.

In the next stage, the influence of the length and permeability of steel strand on stress measurement will be considered comprehensively, and more frequency data will be collected to reduce the error in the experiment. In addition, the relationship between the stress and oscillation frequency of steel strands in concrete structures will also be studied as a key point in order to approach the practical engineering application.

## Figures and Tables

**Figure 1 sensors-19-02782-f001:**
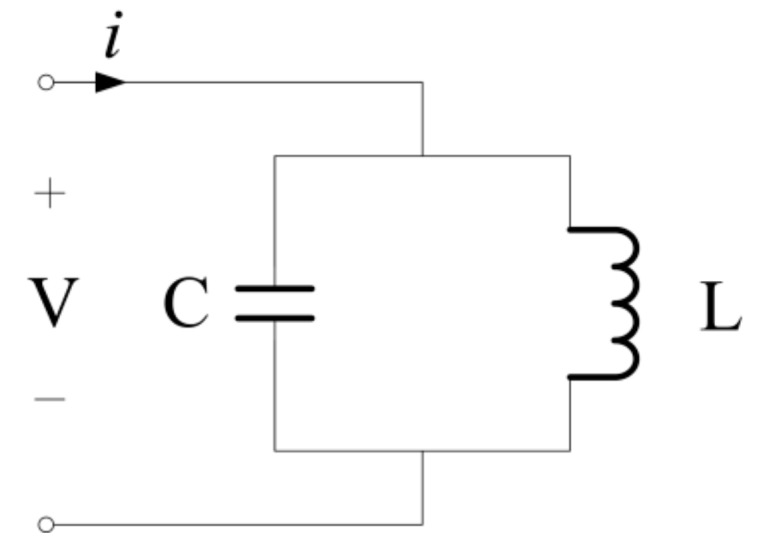
The inductance–capacitance (LC) oscillating circuit.

**Figure 2 sensors-19-02782-f002:**
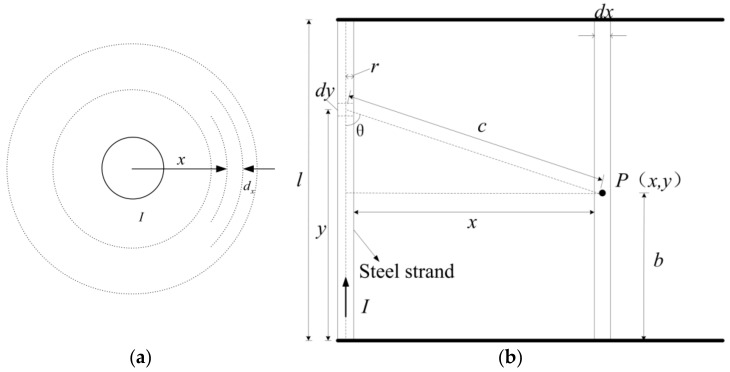
Calculation of external magnetic linkage of segment wire. (**a**) The distribution of external flux linkage; (**b**) Magnetic field model.

**Figure 3 sensors-19-02782-f003:**
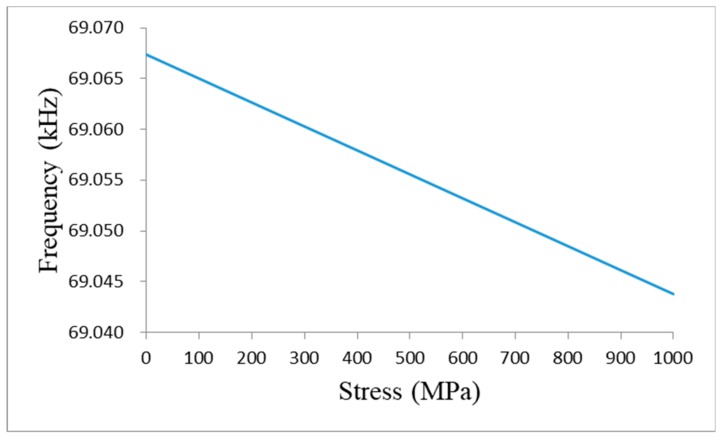
Stress-frequency simulation of the 1.2 m steel strand.

**Figure 4 sensors-19-02782-f004:**
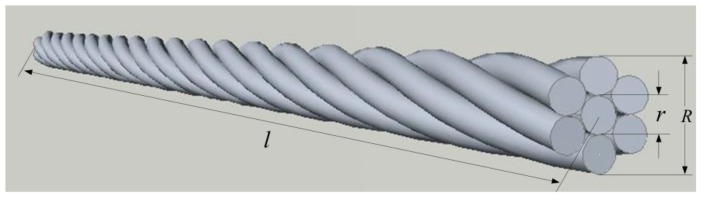
The structure of the steel strand.

**Figure 5 sensors-19-02782-f005:**
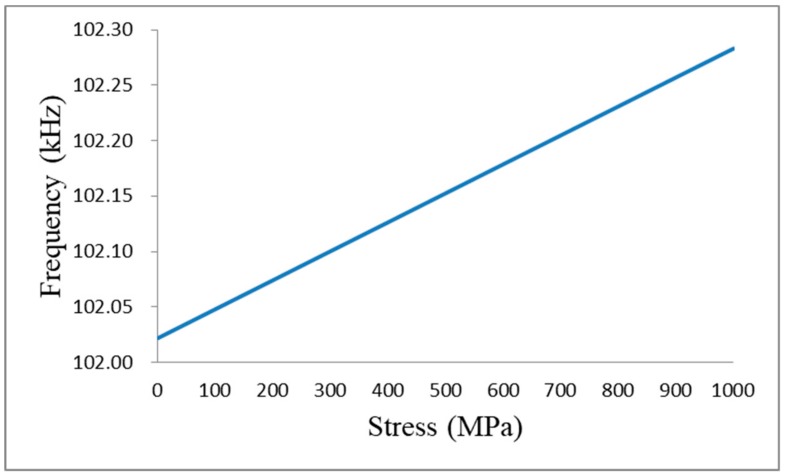
Stress-frequency simulation of the 10 m steel strand.

**Figure 6 sensors-19-02782-f006:**
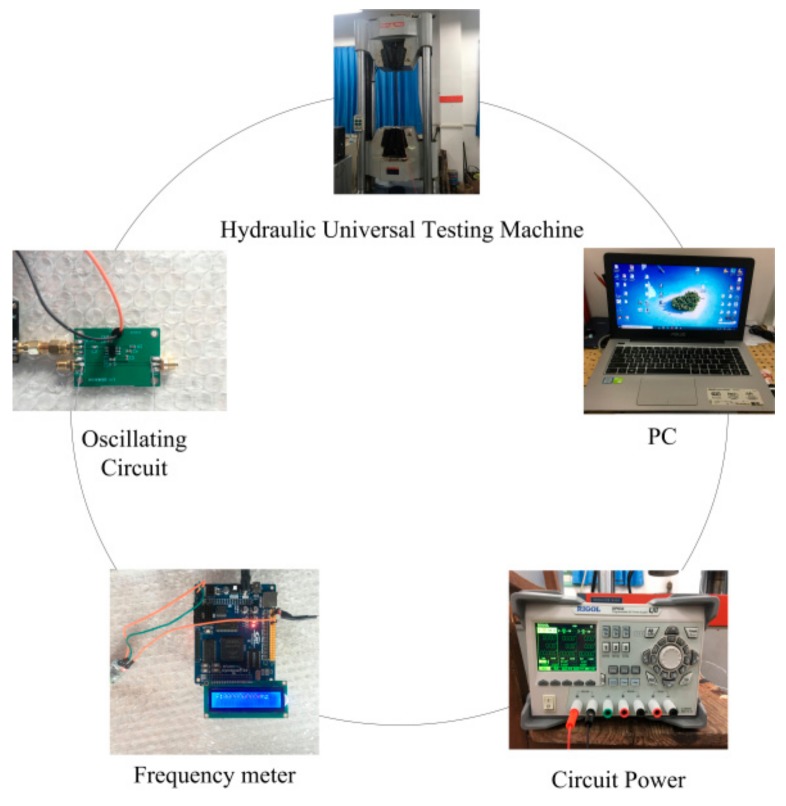
The experimental system of short steel strand.

**Figure 7 sensors-19-02782-f007:**
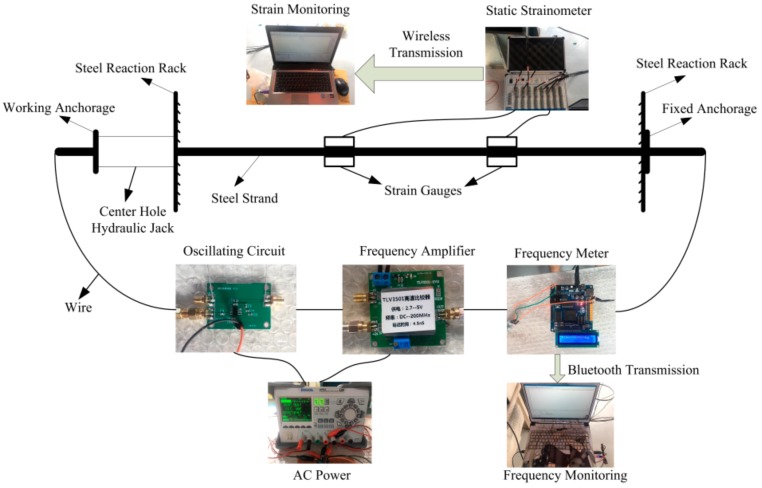
The experimental system of long steel strand.

**Figure 8 sensors-19-02782-f008:**
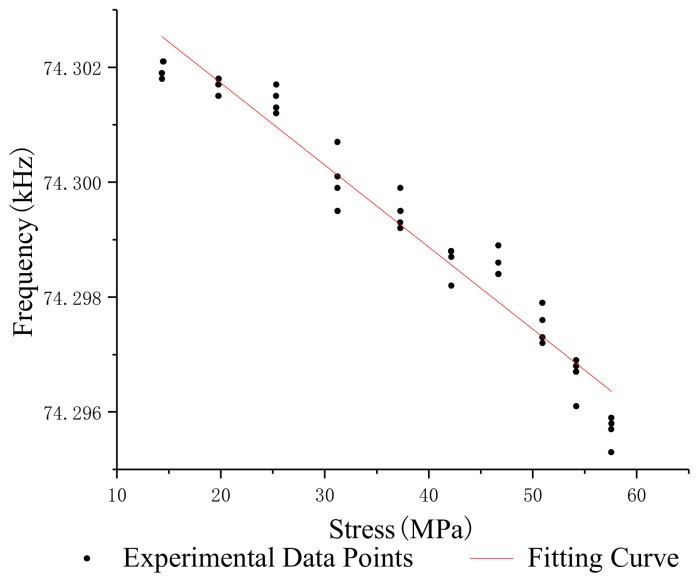
Stress-frequency curve—1.2 m.

**Figure 9 sensors-19-02782-f009:**
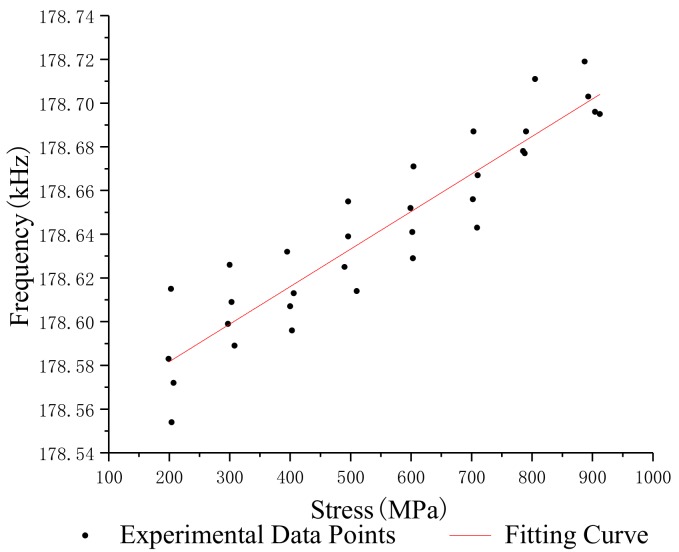
Stress-frequency curve—5 m.

**Figure 10 sensors-19-02782-f010:**
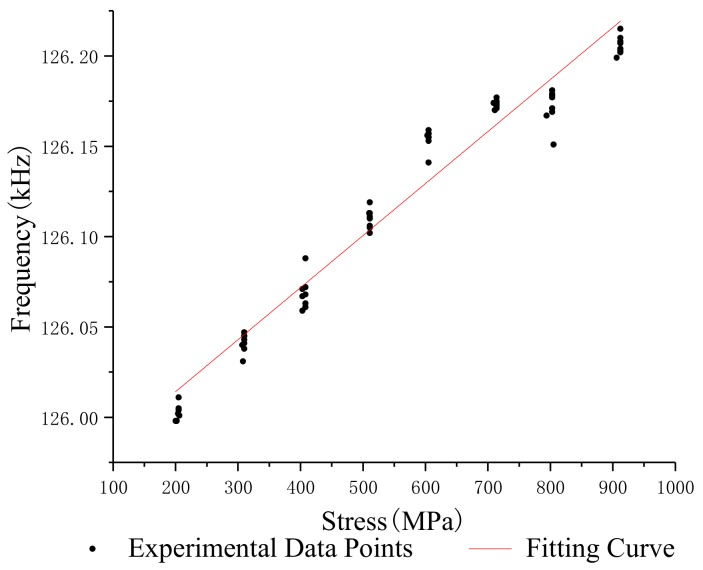
Stress-frequency curve—10 m.

**Figure 11 sensors-19-02782-f011:**
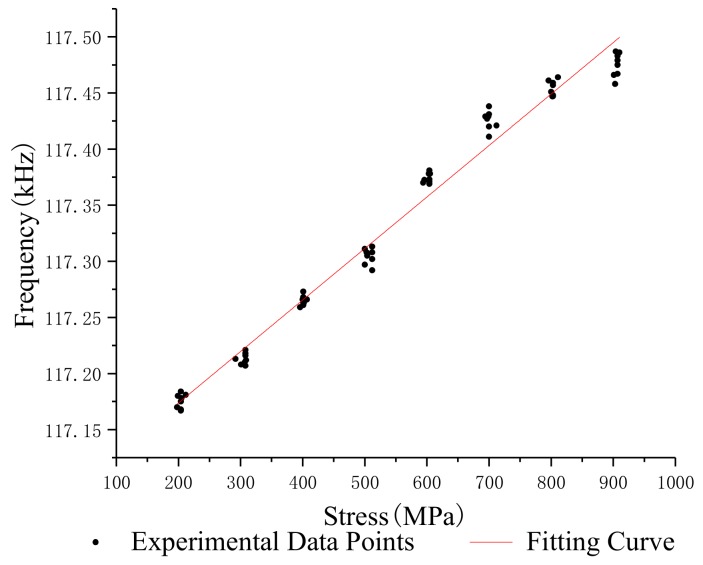
Curve—15 m.

**Figure 12 sensors-19-02782-f012:**
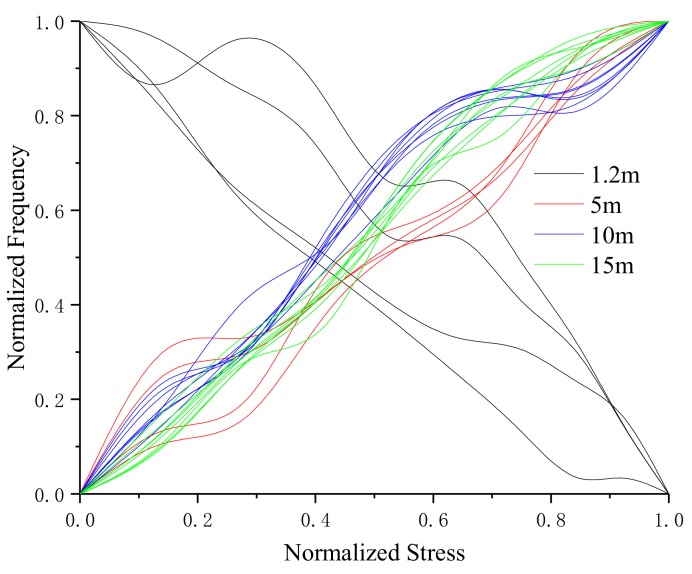
Stress-frequency curve of four steel strands.

**Figure 13 sensors-19-02782-f013:**
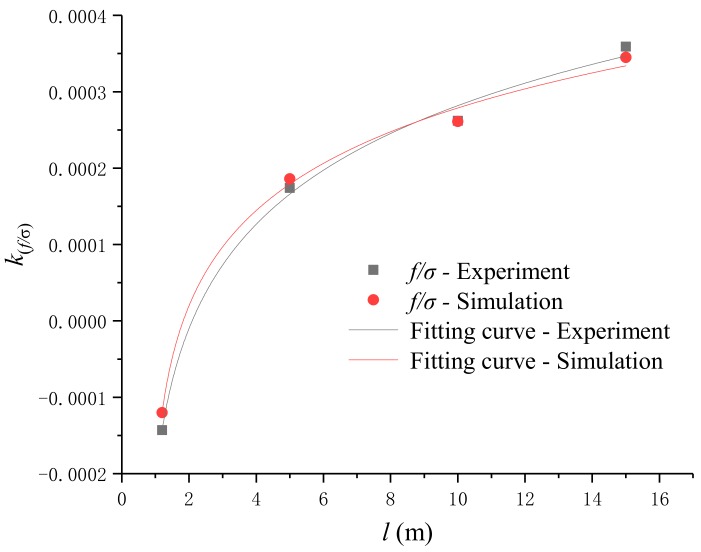
The length effect curves.

**Table 1 sensors-19-02782-t001:** Parameters of steel strands.

Structure of Steel Strand	Length of Steel Strand $l_{0}$ m	Nominal Area of Steel Strand S mm^2^	Nominal Diameter of Steel Strand D mm	Ultimate Tensile Strength R_m_ MPa No Less than	Maximum Tension F_m_ kN No Less than	Maximum Elongation A_GT_ % No Less than
1 × 7	1.2, 5, 10 and 15	139	15.2	1860	260	3.5

**Table 2 sensors-19-02782-t002:** The experimental data—1.2 m.

Average Stress σ/MPa	Measurement Times				SD	RE	MF f/kHz
	Loading 1 f/kHz	Loading 2 f/kHz	Loading 3 f/kHz	Loading 4 f/kHz			
14.389	74.3009	74.2940	74.2748	74.2677	0.01566	0.0211%	74.2844
19.784	74.3018	74.2926	74.2746	74.2669	0.01604	0.0216%	74.2840
25.324	74.3018	74.2911	74.2740	74.2656	0.01636	0.0220%	74.2831
31.223	74.3010	74.2900	74.2733	74.2647	0.01633	0.0220%	74.2823
37.266	74.2995	74.2889	74.2720	74.2639	0.01610	0.0217%	74.2811
42.158	74.2995	74.2883	74.2720	74.2632	0.01626	0.0219%	74.2808
46.691	74.2984	74.2881	74.2712	74.2625	0.01621	0.0218%	74.2801
50.935	74.2972	74.2875	74.2705	74.2619	0.01600	0.0215%	74.2793
54.173	74.2951	74.2869	74.2696	74.2619	0.01528	0.0206%	74.2784
57.554	74.2951	74.2855	74.2687	74.2617	0.01528	0.0206%	74.2778

**Table 3 sensors-19-02782-t003:** The experimental data—5 m.

Average Stress σ/MPa	Measurement Times				SD	RE	MF f/kHz
	Loading 1 f/kHz	Loading 2 f/kHz	Loading 3 f/kHz	Loading 4 f/kHz			
203	178.534	178.572	178.630	178.615	0.04345	0.0243%	178.5878
301	178.589	178.609	178.599	178.626	0.01578	0.0088%	178.6058
402	178.596	178.613	178.607	178.632	0.01508	0.0084%	178.6120
498	178.614	178.625	178.639	178.655	0.01775	0.0099%	178.6333
600	178.629	178.641	178.652	178.671	0.01784	0.0100%	178.6483
706	178.643	178.656	178.667	178.687	0.01863	0.0104%	178.6633
792	178.678	178.677	178.687	178.711	0.01582	0.0089%	178.6883
905	178.695	178.696	178.703	178.719	0.01109	0.0062%	178.7033

**Table 4 sensors-19-02782-t004:** The experimental data—10 m.

Average Stress σ/MPa	Measurement Sequence								SD	RE	MF f/kHz
	Cycle 1 f/kHz Loading	Cycle 1 f/kHz Unloading	Cycle 2 f/kHz Loading	Cycle 2 f/kHz Unloading	Cycle 3 f/kHz Loading	Cycle 3 f/kHz Unloading	Cycle 4 f/kHz Loading	Cycle 4 f/kHz Unloading			
205	125.992	126.005	126.040	126.053	126.108	126.122	126.139	126.155	0.06241	0.0495%	126.0768
310	126.031	126.047	126.049	126.083	126.136	126.133	126.193	126.162	0.05999	0.0476%	126.1043
408	126.087	126.088	126.075	126.087	126.161	126.146	126.228	126.177	0.05543	0.0439%	126.1311
511	126.135	126.112	126.097	126.113	126.190	126.158	126.227	126.185	0.04571	0.0362%	126.1521
605	126.156	126.141	126.119	126.139	126.220	126.229	126.264	126.228	0.05407	0.0429%	126.1870
714	126.174	126.171	126.155	126.196	126.281	126.221	126.298	126.263	0.05474	0.0434%	126.2199
803	126.151	126.169	126.195	126.213	126.267	126.284	126.272	126.284	0.05403	0.0428%	126.2294
912	126.192	126.192	126.219	126.219	126.296	126.296	126.313	126.313	0.05427	0.0430%	126.2550

**Table 5 sensors-19-02782-t005:** The experimental date—15 m.

Average Stres σ/MPa	Measurement Sequence								SD	RE	MF f/kHz
	Cycle 1 f/kHz Loading	Cycle 1 f/kHz Unloading	Cycle 2 f/kHz Loading	Cycle 2 f/kHz Unloading	Cycle 3 f/kHz Loading	Cycle 3 f/kHz Unloading	Cycle 4 f/kHz Loading	Cycle 4 f/kHz Unloading			
204	117.170	117.182	117.190	117.201	117.215	117.219	117.254	117.250	0.03049	0.0260%	117.2101
308	117.204	117.240	117.195	117.219	117.224	117.231	117.289	117.288	0.03524	0.0301%	117.2363
401	117.259	117.314	117.244	117.251	117.253	117.258	117.304	117.321	0.03172	0.0270%	117.2755
512	117.297	117.359	117.278	117.342	117.275	117.315	117.359	117.364	0.03722	0.0317%	117.3236
604	117.370	117.403	117.305	117.389	117.302	117.425	117.401	117.412	0.04747	0.0404%	117.3759
700	117.421	117.438	117.340	117.410	117.408	117.443	117.387	117.438	0.03424	0.0292%	117.4106
803	117.447	117.459	117.403	117.421	117.444	117.478	117.467	117.478	0.02681	0.0228%	117.4496
907	117.471	117.471	117.438	117.438	117.472	117.472	117.491	117.491	0.02038	0.0174%	117.4680

**Notes:** SD is standard deviation, RE is Repeatability error, MF is median frequency.

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
