# Peer review of "Length Effect on the Stress Detection of Prestressed Steel Strands Based on Electromagnetic Oscillation Method"

_sensors, 2019, doi:10.3390/s19122782_

Reviewer 1 Report

Please find it in the attached file.

Author Response

Dear reviewer:

We’d like to sincerely thank you for your careful review. Their valuable suggestions are helpful for improving our paper. We have tried our best to make all the changes suggested by reviewer. Here is a summary of these changes, which are listed one by one as follows. In addition, we have supplied some details in the paper to make it more readable. We sincerely hope that the revised version can satisfy the suggestions and requests.

Please find below our response to the reviews.

 With best regards,

Sincerely yours

Benniu Zhang

 Reviewer 1:

1. In introduction section, the authors should give more details, e.g. advantages and disadvantages, of previous studies to highlight the contribution of the present study.

 Response: Thanks for the reviewer’s valuable suggestion. The authors have described the advantages and disadvantages of previous studies in introduction.(Lines 89 to 91 and lines 100 to 105 of introduction section.)

 2. We have known length will have a significant effect on the stress detection of prestressed steel strands. Why the authors studied this topic again? The authors need to highlight the objective of the study.

 Response: Thanks for the reviewer’s valuable suggestion. Different from other researches, this paper studies the stress measurement of steel strands based on the LC electromagnetic oscillation method. The detection methods and ideas are relatively novel. In this method, only the two ends of the strands that are simulated as inductors are connected to the LC oscillating circuit, and the oscillating frequency of the circuit is measured by a frequency meter to calculate the stress of steel strand. In the previous studies based on the LC electromagnetic oscillation method (Ref. [21-22]), the force-frequency trends of the 1.2m and 10m steel strands are completely opposite. The reason for this phenomenon may be that the length of steel strands are different and the inductance characteristics are different. Therefore, this paper carries out related theoretical and experimental research on steel strands of different lengths. A supplementary explanation has been added in section 1.3.

 3. In Figs. 10-12, the results for the 1.2m strand present an opposite trend with the results for the 5, 10 and 15 strands. The authors may give the explanation (mechanism).

 Response: Thanks for the reviewer’s suggestion. Should Figs. 10-12 be Figs. 9-11? Because the 1.2m steel strands mainly exhibits the inductance characteristics of segment wire, while the 5m, 10m and 15m steel strands mainly exhibit the inductance characteristics of the spiral coil. When the strand shows the inductance characteristic of segment wire, the stress increases, the inductance increases, and the frequency decreases. When the strand shows the inductance characteristic of the conductor, the stress increases, the inductance decreases, and the frequency increases. Therefore, the data analysis of the tensile test of 1.2 m steel strand shows that the frequency decreases with the increase of stress; while the test analysis of 5 m, 10 m and 15 m steel strands shows that the frequency increases with the increase of stress. Additional explanations have been made in section 3.3.(Lines 375 to Line 377.)

 4. Why the authors choose 1.2, 5, 10 and 15 m strands for study? The reviewer does not believe the four cases are enough to obtain an empirical model. 2.013 and 2.019m are questionable. More cases are recommended to verify the conclusion.

 Response: Thanks for the reviewer’s suggestion. It may be that the authors do not describe clearly, the critical length of steel strand is (2.013, 2.199). More steel strands of different lengths should be selected as test samples, but due to the limitation of test conditions, 2 m, 3 m or 4 m strands can not be fixed on test equipment. In addition, 1 m, 5 m, 10 m and 15 m steel strands should be selected as test samples, but the limitation of the universal test machine, only 1.2 m that fits test equipment can be selected. After the author's analysis and calculation, the length and number of test samples will not affect the final results.

 5. The reviewer is confused with the result(1.2, -0.000120). Both frequency and stress are positive, how to get the negative value? The authors should explain.

 Response: This suggestion is valuable. The meaning of / σ is slope k(/ σ), which is caused by the authors’ carelessness. Figure 13, Equations (31), (32) and contents of section 4.1 have been all modified in the manuscript.

 6. Please shorten the conclusion section.

 Response: Thanks for the reviewer’s suggestion. We have shortened the conclusion section in the manuscript.

 7. Line 344 “the median filter... ” The authors may explain why they use median values in analysis.

 Response: In order to improve the measurement accuracy of the oscillation frequency, the output data detected by the frequency meter are the average value of multiple measurements within the set time, that is, the median frequency used for data analysis in the manuscript. This method has two advantages in data processing: a) Reducing the workload of data processing; and b) Improving the accuracy of frequency data.

 8. Should the last sentence “Therefore, this paper ... LC electromagnetic vibration method for stress detection of steel strands." in the introduction be “Therefore, this paper ... LC electromagnetic oscillation method for stress detection of steel strands." (1.3. Electromagnetic Oscillation Method). Please confirm it.

 Response: Thanks for the reviewer’s suggestion. The expression of “LC electromagnetic vibration method” should be “LC electromagnetic oscillation method”, The authors have revised it in this manuscript.

 9. The application of the electromagnetic oscillation method to stress detection of      prestressed steel strand is potentially an important demonstration of the technology. It would be interesting to know whether this method is suitable for stress detection of steel strands in concrete structures?

 Response: The reviewer's suggestion is valuable. In the next stage, we will try to apply this method to the stress detection of steel strands in concrete structures.

 10. There are typos and grammatical errors. For example, " the experimental system is composed as shown in Figure 6.' can be 'the experimental system is presented in Figure 6". "3.2.2 The experimental devices and procedure of short steel strand" should be "3.2.2 Experimental devices and procedure of short steel strand". A careful proofreading is recommended.

 Response: Thank the reviewer very much for your suggestion. Now, we have improved the organization of the manuscript and made some revisions. We’ve tried our best to solve the grammar and typos problem.

 Thanks again for your comments and suggestions.

Reviewer 2 Report

Detection of prestress loss in PC structure is an important research topic. This manuscript presented a LC electromagnetic oscillation-based method for stress detection of prestressed steel strands considering length effect, which is interesting. My main comments lie in:

(1) Please clarify the differences among this manuscript with https://doi.org/10.1155/2018/1584903 and

https://doi.org/10.1016/j.measurement.2018.05.014. If the results obtained in this manuscript cover these in the above two papers, please explain the rationality.

(2) Line 267: “The stress-frequency simulation result of 10 m steel strand is shown in Figure 4” should be Figure 5.

(3) Please explain the consistency of results between Figure 5 (Theoretical one) and Figure 10 (Experimental one). How do the authors guarantee the accuracy and consistency of theoretical and experimental results.

(4) Taking the results shown in Figure 11 for example, the frequency increases from (117.15-117.20)kHz to (117.45-117.50)kHz, when stress increases from 200MPa to 900MPa. The absolute change of frequency is 0.3kHz, and relative change ratio is below 3‰, how to guarantee the testing accuracy in real application.

(5) Most importantly, detection of stress for prestressed strands is relatively simple. The application of LC electromagnetic oscillation method for prestress loss detection through considering the coupling effect of concrete and prestress strands should be added to verify the applicability of EMO method.  For example: https://www.mdpi.com/1424-8220/16/8/1317/htm

Author Response

Dear reviewer:

We’d like to sincerely thank you for your careful review. Their valuable suggestions are helpful for improving our paper. We have tried our best to make all the changes suggested by reviewer. Here is a summary of these changes, which are listed one by one as follows. In addition, we have supplied some details in the paper to make it more readable. We sincerely hope that the revised version can satisfy the suggestions and requests.

Please find below our response to the reviews.

 With best regards,

Sincerely yours

Benniu Zhang

 Reviewer 2:

1. Please clarify the differences among this manuscript with https://doi.org/10.1155/2018/1584903 and https://doi.org/10.1016/j.measurement.2018.05.014. If the results obtained in this manuscript cover these in the above two papers, please explain the rationality.

Response: There are two differences between the two published articles and this manuscript: a) The published papers (Ref. [21-22]) only focus on the force-frequency variation trend of 1.2 m and 10 m steel strands. They are the validation of the feasibility of LC electromagnetic oscillation method, and do not explain the reason why the force-frequency variation is positively (negatively) correlated. While this manuscript explains the reasons for the opposite trend of 1.2m and 10m steel strands, and establishes relevant theoretical models; and b) This manuscript carries out experimental research on 1.2 m, 5 m, 10 m and 15 m steel strands, and obtains new experimental data to prove the rationality of the theoretical model in this paper. The critical interval for distinguishing long and short steel strands is also obtained through analysis. The contents have been supplemented in section 1.3.

2. Line 267: “The stress-frequency simulation result of 10 m steel strand is shown in Figure 4” should be Figure 5.

Response: Thanks. We have revised it in the manuscript (Line 276).

3. Please explain the consistency of results between Figure 5 (Theoretical one) and Figure 10 (Experimental one). How do the authors guarantee the accuracy and consistency of theoretical and experimental results.

Response: The reviewer's suggestion is valuable. The consistency between Figure 5 (Theoretical one) and Figure 10 (Experimental one) can be expressed as following:a) The oscillation frequency increases with the increase of stress; b) The stress of 10m steel strand in Figure 5 increases from 200 MPa to 900 MPa, the oscillation frequency increases by 0.1828kHz, and the oscillation frequency of 10m steel strand in Figure 10 increases by 0.1782kHz. However, the authors cannot guarantee the accuracy and consistency of theoretical and experimental results. The theoretical derivation in this manuscript is based on the experimental results of the previous studies (Ref. [21-22]), and then the new experiment is implemented to verify the rationality of the theoretical model. The effects of factors such as length of steel strand and external magnetic field are considered in this study, but if there are new influencing factors, the accuracy and consistency of the theoretical and experimental results cannot be guaranteed.

4. Taking the results shown in Figure 11 for example, the frequency increases from (117.15-117.20) kHz to (117.45-117.50) kHz, when stress increases from 200MPa to 900MPa. The absolute change of frequency is 0.3kHz, and relative change ratio is below 3‰, how to guarantee the testing accuracy in real application.

Response: In the experimental system, the measurement accuracy of the frequency meter is 0.0001 kHz. The stress increases from 200 MPa to 900 MPa, and the frequency varies by 0.3 kHz, which is 3,000 times the accuracy of the frequency meter. Therefore, it can completely fit the requirements of measurement. The system will be disturbed by some external factors, so we can not guarantee the accuracy of measurement in concrete structures. Solving the interference of stress measurement is what the authors will do in the future.

5. Most importantly, detection of stress for prestressed strands is relatively simple. The application of LC electromagnetic oscillation method for prestress loss detection through considering the coupling effect of concrete and prestress strands should be added to verify the applicability of EMO method.  For example: https://www.mdpi.com/1424-8220/16/8/1317/htm.

Response: For now, we mainly focus on the research of steel strand, and the analysis of the coupling effect between steel strands and concrete will be carried out in the next stage.

 Thanks again for your comments and suggestions.

Reviewer 3 Report

The paper is related to application of electromagnetic oscillation method for prestress detection of structures. The paper contains both theoretical and experimental investigations.

 Remarks:

1) What is an accuracy of frequency determination during experimental tests? Due to lack of such information it is hard to see applicability of the proposed method. Maximal differences among frequencies are related to the first decimal place of the frequency values.

2) Theoretical analyses are confirmed by sentences related to previous papers of the authors. Percentage differences or other indicators should be added to show what is a relationship between theoretical and experimental analyses.

3) How the method can be applied in practice? Is it possible having frequency values determine prestress value of structure?

4) Differences among the paper and previously published ones [21], [22] should be highlighted.

5) Table 2-5. Symbols: SD, RE, MF should be explained.

The paper is not recommended for publication in the presented form.

Author Response

Dear reviewer:

We’d like to sincerely thank you for your careful review. Their valuable suggestions are helpful for improving our paper. We have tried our best to make all the changes suggested by reviewer. Here is a summary of these changes, which are listed one by one as follows. In addition, we have supplied some details in the paper to make it more readable. We sincerely hope that the revised version can satisfy the suggestions and requests.

Please find below our response to the reviews.

 With best regards,

Sincerely yours

Benniu Zhang

Reviewer 3:

1. What is an accuracy of frequency determination during experimental tests? Due to lack of such information it is hard to see applicability of the proposed method. Maximal differences among frequencies are related to the first decimal place of the frequency values.

Response: The reviewer's suggestion is valuable.The accuracy of the frequency meter in the measurement system is 0.0001kHz. The relevant information has been added in this manuscript.(Lines 308 to 309.)

2. Theoretical analyses are confirmed by sentences related to previous papers of the authors. Percentage differences or other indicators should be added to show what is a relationship between theoretical and experimental analyses.

Response: It may be that the author did not elaborate clearly. The error of theoretical and experimental analysis has been given in Section 4.2 (Lines 452 to 453).

3. How the method can be applied in practice? Is it possible having frequency values determine prestress value of structure?

Response: For now, we mainly focus on the research of steel strand, and the analysis of the coupling effect between steel strands and concrete will be carried out in the next stage.

4. Differences among the paper and previously published ones [21], [22] should be highlighted.

Response: Differences among the paper and previously published ones [21], [22] can be summarized as follows:a) The published papers (Ref. [21-22]) only focus on the force-frequency variation trend of 1.2 m and 10 m steel strands. They are the validation of the feasibility of LC electromagnetic oscillation method, and do not explain the reason why the force-frequency variation is positively (negatively) correlated. While this manuscript explains the reasons for the opposite trend of 1.2m and 10m steel strands, and establishes relevant theoretical models. b) This manuscript carries out experimental research on 1.2 m, 5 m, 10 m and 15 m steel strands, and obtains new experimental data to prove the rationality of the theoretical model in this paper. The critical interval for distinguishing long and short steel strands is also obtained through analysis. The contents have been supplemented in section 1.3.

 5. Table 2-5. Symbols: SD, RE, MF should be explained.

Response: It may be that the author did not elaborate clearly. Actually, these symbols, for instance, SD, RE, MF are all explained in this manuscript(Line 364).

 Thanks again for your comments and suggestions.

Round  2

Reviewer 2 Report

This manuscript has addressed the comments, which can be accepted.